# Degradation of Bisphenol A by CeCu Oxide Catalyst in Catalytic Wet Peroxide Oxidation: Efficiency, Stability, and Mechanism

**DOI:** 10.3390/ijerph16234675

**Published:** 2019-11-23

**Authors:** Zhaojie Jiao, Ligong Chen, Guilin Zhou, Haifeng Gong, Xianming Zhang, Yunqi Liu, Xu Gao

**Affiliations:** 1Engineering Research Center for Waste Oil Recovery Technology and Equipment of Ministry of Education, Chongqing Technology and Business University, Chongqing 400067, China; ligongchen2019@163.com (L.C.); dicpglzhou@ctbu.edu.cn (G.Z.); ghfpy@ctbu.edu.cn (H.G.); zxm215@126.com (X.Z.); 2State Key Laboratory of Heavy Oil Processing, China University of Petroleum (East China), Qingdao 266580, China; liuyq@upc.edu.cn; 3National Research Base of Intelligent Manufacturing Service, Chongqing Technology and Business University, Chongqing 400067, China

**Keywords:** catalytic wet peroxide oxidation, CeCu oxide catalyst, bisphenol A (BPA), degradation pathway

## Abstract

The CeCu oxide catalyst CC450 was prepared by citric acid complex method and the catalytic wet peroxide oxidation (CWPO) reaction system was established with bisphenol A (BPA) as the target pollutant. By means of characterization, this research investigated the phase structure, surface morphology, reducibility, surface element composition, and valence of the catalyst before and after reuse. The effects of catalyst dosage and pH on the removal efficiency of BPA were also investigated. Five reuse experiments were carried out to investigate the reusability of the catalyst. In addition, this research delved into the changes of pH value, hydroxyl radical concentration, and ultraviolet-visible spectra of BPA in CWPO reaction system. The possible intermediate products were analyzed by gas chromatography-mass spectrometry (GC-MS). The catalytic mechanism and degradation pathway were also discussed. The results showed that after reaction of 65 min, the removal of BPA and total organic carbon (TOC) could reach 87.6% and 77.9%, respectively. The catalyst showed strong pH adaptability and had high removal efficiency of BPA in the range of pH 1.6–7.9. After five reuses, the removal of BPA remained above 86.7%, with the structure of the catalyst remaining stable to a large extent. With the reaction proceeding, the pH value of the reaction solution increased, the concentration of OH radicals decreased, and the ultraviolet-visible spectrum of BPA shifted to the short wavelength direction, that is, the blue shift direction. The catalysts degraded BPA rapidly in CWPO reaction system and the C–C bond or O–H bond in BPA could be destroyed in a very short time. Also, there may have been two main degradation paths of phenol and ketone.

## 1. Introduction

Bisphenol A (BPA) is one of the highest production-volume chemicals in the world. The global production of BPA reached 8 million metric tons in 2016 and is expected to reach 10.6 million metric tons by 2022 [1]. It is mainly used as an intermediate chemical in the manufacturing of polycarbonate and epoxy resin, unsaturated polyester styrene resin, and flame retardant. It is also used in the preparation of antioxidants, stabilizers, plasticizers, pot coatings, powder coatings, heat-sensitive paper additives, dented aluminum fillers, and insecticides, among others [2,3]. It mainly affects the environment through the production and processing of BPA-containing raw materials, the treatment of incomplete wastewater, landfill leachate, and the leaching of BPA-containing waste. BPA has been detected in lakes, rivers, drinking water, food, beverages, soil, and air [4,5]. BPA was identified as an endocrine disrupting chemical (EDC) by the United States Environmental Protection Agency (EPA) and the World Wide Fund for Nature (WWF). BPA has biological accumulation, which is relatively stable in the environment. The high hydrophobicity of BPA makes it easy to deposit in sludge after draining into water, and the half-life of BPA in anaerobic sediment is very long, about 70 days [6]. It has shown high toxicity to freshwater and marine species in aquatic organisms in the range of 1000–10,000 μg L^−1^ [3]. A small amount of ingestion can destroy the endocrine system of the human body, which may cause infertility, fetal abnormalities, breast cancer, and so on, and have potential harm to the safety of the ecosystem and human health [7,8,9]. Therefore, much attention should be paid to the treatment of BPA wastewater.

In recent years, a variety of methods including physical, biological, and chemical methods have been developed to degrade BPA in aqueous solution, for example, adsorption [10,11], biological degradation [8,12] and photocatalysis [13,14], ozonation [15], ultrasonication [16,17], electrocatalytic oxidation [18], Fenton and Fenton-like processes [19], and catalytic wet air oxidation [20]. Physical methods such as activated carbon adsorption can transfer pollutants from one medium to another, yet still pose a potential threat to the environment. Biodegradation methods often take a long time, as they may be affected by the toxicity of BPA and high concentration of refractory organic matter. To a large extent, they also depend on many environmental factors, such as nutrition, temperature, and salinity. Among these methods, advanced oxidation processes (AOPs) are considered as effective and promising methods to destroy persistent organic pollutants (POPs). Among the advanced oxidation methods, catalytic wet peroxide oxidation (CWPO) uses heterogeneous catalysts instead of homogeneous catalysts, which reduces the loss of active components and improves the reusability of catalysts. Therefore, it has greater advantages. To date, various catalysts containing transition metals, rare earth metals, or precious metals have been used to degrade BPA in the CWPO process. These catalysts are all reported to exhibit excellent BPA degradation properties, and Table 1 shows the main results from these reports [21,22,23,24,25,26,27,28,29]. These catalysts also have some shortcomings, for instance, they should contain irons, pH value of the reaction should be around 3 [21,22,23], and the catalyst lacks stability. For example, using the Fe-C (GS) catalyst to degrade BPA, there would be about a 20% loss of Fe in the early stages [21]. Using the Fe-Mt-TC-C (Iron-pillared montmorillonite Tetracycline carbon) catalyst to degrade BPA, the efficiency decreased from 100% to 17% when the catalyst was recycled for three times [22]. Although the catalysts containing precious metal will have a good catalytic effect on the degradation of BPA, they are not the first choice in the degradation of refractory organics [29] considering their high prices. Materials containing Cu^2+^/Cu^+^, such as iron and other metals, could produce hydroxyl radical in catalyzing hydrogen peroxide, and also have a higher ability to degrade BPA [25]. However, the Cu-based catalyst leads to the loss of active components in the CWPO process, and the preparation process of some catalysts is relatively complicated [24,26,30]. Therefore, this paper intends to prepare a catalyst with simple structure and strong stability, which could also show high catalytic activity and adapt to wide pH ranges.

In this paper, a simple CeCu mixed oxide catalyst was prepared through citric acid-assisted complexation method and characterized. The CWPO results were analyzed in terms of BPA (or total organic carbon (TOC)) removal and H_2_O_2_ decomposition. This paper investigated the stability and reusability of the catalyst, and analyzed the possible intermediate products in CWPO system by investigating the changes of acidity and alkalinity, hydroxyl radical concentration, and ultraviolet-visible spectrum of pollutants in the reaction process, combined with gas chromatography-mass spectrometry detection. The pathway and mechanism of catalytic oxidation degradation were discussed.

## 2. Experimental

### 2.1. Materials

BPA was purchased from Adamas Reagent Co., Ltd. H_2_O_2_ (30%, *w*/*w*) and citric acid (CA) were purchased from Chongqing East Chemical Group Co., Ltd. Cu(NO_3_)_2_·3H_2_O and Ce(NO_3_)_3_·6H_2_O were purchased from Chengdu Kelong Chemical Reagent Factory, China. All the chemicals were analytical pure grade without any further pre-treatment. Deionized water was used as the experimental water. 

### 2.2. Preparation of Catalyst 

In a typical preparation process, Cu(NO_3_)_2_·3H_2_O, Ce(NO_3_)_3_·6H_2_O, and CA (Ce/Cu molar ratio = 1.0; CA/(Ce + Cu) molar ratio = 1.8) were dissolved in a certain amount of distilled water [27]. The mixture was then stirred at 80 °C until the water completely evaporated. Afterward, the sample was dried at 100 °C for 20 h, and then the dried sample was heated to 450 °C in a muffle furnace at 10 °C min^−1^, and calcined at 450 °C for 3.0 h to obtain a CeCu oxide catalyst that was named CC450. The initial molar ratio of copper to cerium was 1.08 by an inductively coupled plasma optical emission spectroscopy (ICP-OES) analysis.

### 2.3. Characterizations of the Catalyst

The X-ray diffraction (XRD) studies of CeCu oxide catalysts were performed as 2θ ranged from 20° to 80° on a Rigaku XRD-6100 X-ray diffractometer with Cu Kα linefiltered by Ni. The tube had 40 kV voltage and 30 mA current. The scanning rate was 5° min^−1^. The Scanning electron microscopy (SEM) images of CeCu-mixed oxide catalysts were captured with a Hitachi’s S-4800 instrument. In each H_2_ temperature programmed reduction (H_2_-TPR) test of CeCu-mixed oxide catalysts, sample in 30 mg was placed in a U-shape quartz tube. The flow (25 mL min^−1^) of 5.0% H_2_-Ar mixture gas was controlled by a mass flow controller with the ramping rate of 10 °C min^−1^. The X-ray photoelectron spectroscopy (XPS) measurements were employed to determine the chemical states of Ce, Cu, and O in the studied CeCu oxide catalysts. The signals were collected by a KRATOS X-ray source (model XSAM800) with an aluminium crystal, operating at 12 kV anode voltages and 12 mA emission current. 

### 2.4. Catalytic Degradation Experiments and Analytical Methods

The dosage of catalyst and hydrogen peroxide, the initial pH of the solution, the concentration of BPA, and the reaction temperature all had a certain influence on the removal of BPA. This experiment mainly investigated the catalyst dosage effect and initial pH of the solution on the removal of BPA. The degradation of BPA was carried out in a 100 mL conical flask. In a typical process, 1 g L^−1^ of catalyst was added into 25 mL of 152 mg L^−1^ of aqueous solution of BPA; the initial pH value of the solution was 6.6, then followed by the addition of 196 mmol L^−1^ of H_2_O_2_ (30 wt. %), after which it was quickly placed the conical flask in a water bath in which the temperature was maintained at 75 ± 0.5 °C. After the reaction, the reaction solution was centrifuged at 6000 rpm for 5 min. The supernatant fraction was extracted to measure BPA concentration and the TOC value. 

The concentration of BPA was determined by ultraviolet-visible spectrophotometer (UV-2550, Shimadzu, Kyoto, Japan) at a wavelength of 274.6 nm. The TOC was determined by TOC analyzer (TOC-VCPN, Shimadzu, Kyoto, Japan). The concentration of H_2_O_2_ was determined by using a titanium sulfate spectrophotometric method [21]. The initial copper and cerium content of the prepared catalytic material and the leaching concentration of Cu^2+^ was determined by ICP-OES (ICP2060T, Jiangsu Skyray Instrument Co., Ltd., Kunshan, China). The degradation intermediate products of BPA were qualitatively analyzed by gas chromatography-mass spectrometry (GC-MS, 7890b/5977b, Agilent, CA, USA).

GC-MS intermediate analysis: instrument conditions, chromatographic column, HP-5 (0.25 μm, 30 m × 0.25 mm); inlet temperature, 260 °C; carrier gas, He; flow rate, 1 mL min^−1^; split ratio: 10/1; injection volume: 1 μL. Temperature rising procedure: column temperature 40 °C, held for 3 min, then increased at 15 °C min^−1^ to 100 °C; held for 2 min, then increased at 15 °C min^−1^ to 180 °C; held for 2 min, then increased at 15 °C min^−1^ to 240 °C; held for 2 min, then increased at 15 °C min^−1^ to 280 °C and held for 2 min. MS conditions: quadrupole temperature, 150 °C; ion source, 230 °C; electron bombardment source, 70 eV; mass scanning range, 45–280 M/Z. 

### 2.5. Quality Control

ICP-OES has become a mature technology for trace analysis of biological and environmental samples [31,32]. The ICP2060T is capable of analyzing sample concentrations from a few part per billion (PPb) to a few percent or even tens of percent. The concentrations of Cu^2+^ and Ce^4+^ measured in this experiment were greater than 1ppm; therefore, this method can be used to detect the concentration of ions in solution, which can meet the accuracy requirements. The operation conditions for the ICP-OES, as well as the values of the limit of detection (LOD), limit of quantification (LOQ), and correlation factor of linear curve for the analytes Cu and Ce are summarized in Table 2. The curves were constructed with one calibration blank and five calibration standards. The calibration standards consisted of solutions with concentrations of 1, 2, 3, 4, and 5 ppm for each element. Cu and Ce use the reference materials of China Guobiao (Beijing) Testing and Certification Co., Ltd. For quality assurance and control, the 3 ppm standard was used as an instrument performance check solution. Prior to each sample test, the standard curve was calibrated with the standard solution. Moreover, each sample was measured three times and averaged. 

## 3. Results and Discussion

### 3.1. Characterizations of the Catalysts

Figure 1 shows the XRD patterns of the catalyst before and after reuse. It demonstrates that after reusing the catalyst for five times, high intensity CeO_2_ crystal phase diffraction peaks were formed at 28.5°{111}, 33.1°{200}, 47.5°{220}, and 56.4°{311}, and weak intensity CuO crystal phase diffraction peaks were formed at 35.6° and 38.7°. These were consistent with the position of the crystal phase diffraction peak of the catalyst before the reaction. It can be seen that the crystal structure of the catalyst did not change obviously after five cycles of use, but the peak strength of the species changed greatly, which indicated that the prepared CeCu oxide catalyst was slightly affected by BPA in heterogeneous CWPO reaction and, to a large extent, it could maintain a good structural stability.

SEM was used to analyze the morphology of the catalyst after five reuses. The results are shown in Figure 2. From the comparison figures before and after the reaction, it was seen that the catalyst had a lot of rough, loose, and porous flocculent structure. After reuse, no obvious morphological changes were observed, such as formation of junctions and disappearance of pore structure. The results show that the catalysts have good mechanical stability.

The crystalline phase structure, pore structure, and surface morphology of the catalyst did not change significantly before and after the reaction. On the one hand, it may have been due to the fact that Cu^2+^ dissolved into the CeO_2_ lattice to form CeCu oxide solid solution, which resulted in strong interaction between CuO and CeO_2_ and enhanced the stability of the catalyst; on the other hand, it may be due to the addition of rare earth element Ce, which can disperse and stabilize the active components of the catalyst and improve its acidity resistance and prevent volume shrinkage, thereby improving the structural stability of the catalyst to a certain extent [30].

When the catalyst was reused for many times, the composition of the catalyst could be reflected by investigating its reducibility. Figure 3 showed that after five reuses, the catalysts formed hydrogen consumption peaks of α and β at 220 and 262 °C, respectively. Compared with the hydrogen consumption peaks of α and β formed at 140 and 200 °C before the reaction, the position of hydrogen consumption peaks shifted to high temperature. Concurrently, the initial reaction temperature (shown by the black arrow in the Figure 3) also moved to a high temperature after reuse. In addition, the peak strength and peak area decreased significantly. The above results indicated that the reducibility of catalyst decreased significantly. This may have been mainly due to the fact that as the reuse times of the catalyst increased, the CuO of the amorphous phase partially dispersed on the surface of the catalyst and was likely to fall off and dissolve in water. As a result, the amount of species that can react with hydrogen decreased, thus leading to a decrease in reducibility.

Figure 4 shows the XPS spectra of the catalyst before and after five reuses. Figure 4a shows that the Ce3d XPS spectrum of the CeCu oxide catalyst after five reuses showed six obvious XPS peaks at electron binding energies 916.6, 907.1, 900.5, 898.9, 888.9, and 882.3 eV. Among them, the peaks at electron binding energies 900.5 eV and 882.3 eV belonged to the main peaks of Ce3d_3/2_ and Ce3d_5/2_, respectively. The peaks at 916.5, 907.1, 897.9, and 888.2 eV refer to satellite peaks of Ce 3d_3/2_ and Ce 3d_5/2_. These peak positions and shapes were the same as the Ce species on the surface of the catalyst prepared before the reaction.

From Figure 4b, it can be seen that the Cu2p XPS spectra of CeCu oxide catalysts formed four distinct XPS peaks at electronic binding energies 961.6, 953.6, 941.8, and 933.6 eV after five reuses. The peaks at 953.6 eV and 933.6 eV belonged to the main peaks of Cu2p_1/2_ and Cu2p_3/2_, respectively. The peaks at 961.6 eV and 941.8 eV belonged to satellite peaks of Cu 2p1/2 and Cu 2p3/2, respectively. These peak positions and shapes were also the same as the Cu species on the surface of the catalyst prepared before the reaction.

From Figure 4c, the O1s XPS spectra of CeCu oxide catalysts after five reuses formed three XPS peaks at electron binding energies 531.6, 530.3, and 529.2 eV, respectively. The peaks ranging from 528.5–529.8 eV belonged to lattice oxygen O^2−^ species on the catalyst surface. The peaks ranging from 529.8–531 eV belonged to the absorption oxygen O_2_^2−^ or O^−^ species on the catalyst surface. The peak at 531.6 eV was from the hydroxyl oxygen OH^−^ [30] of absorbed water on catalyst surface. From the comparison of O1s XPS spectra before and after the reaction, the O1s XPS peak of CC450 catalyst shifted slightly to the right after five reuses, indicating that lattice oxygen content was gradually dominant and stable crystal phase CuO might gradually form on the catalyst surface, with some loose amorphous species lost along with the reaction.

XPS analysis showed that the composition and valence of elements on the surface of the catalyst did not change significantly after five times of reuse, but the content changed, that is, some loose amorphous components may have been lost and the structure gradually stabilized.

### 3.2. Catalytic Performances of the Catalysts

Before the experiment, the effect of single factor on the removal of BPA simulated wastewater was studied. The results are shown in Figure 5, which shows that after 95 min of reaction, the contribution of volatilization of BPA-simulated wastewater, single adsorption of catalyst, and single oxidation of H_2_O_2_ to BPA removal was 7.14, 9.02, and 7.46%, respectively. These factors were relatively low, indicating that the influence of these factors was secondary and negligible under the condition of heterogeneous CWPO reaction. When the cc450 catalyst was completely dissolved in 25 mL of the reaction solution by 3% nitric acid, the concentrations of Cu^2+^ and Ce^4+^ were 153.2 and 311.7 mg L^−1^, respectively, by ICP-OES analysis. 

Figure 6 shows that both the removal of BPA and the consumption of H_2_O_2_ increased with the increase of catalyst dosage. After 15 min of reaction, when the dosage of catalyst increased from 0.2 g L^−1^ to 1.4 g L^−1^, the removal of BPA increased from 36.2% to 63.7%, and the consumption of hydrogen peroxide increased from 32.2% to 69.9%. After 65 min of reaction, the removal of BPA increased from 70.9% to 89.3%, and the consumption of H_2_O_2_ increased from 62.3% to 99.9%. There was a big difference in BPA removal between the two time periods, which indicated that the dosage of catalyst had a relatively large impact on BPA removal. Considering the removal of BPA and the consumption of H_2_O_2_ in the whole reaction process, the dosage of catalyst 1 g L^−1^ was reasonable. After 65 min of reaction, the removal of BPA and TOC were 87.6% and 77.9%, respectively, and the leaching concentration of Cu^2+^ was 26.3 mg L^−1^.

In the absence of catalyst, the removal of BPA was very low, which indicated that it was difficult for a single H_2_O_2_ to achieve catalytic oxidation degradation of BPA. With the addition of catalyst, the removal of BPA increased significantly, which indicated that in the heterogeneous CWPO reaction, the addition of catalyst effectively promoted the decomposition of hydrogen peroxide and formed strong oxidation ·OH radical, which changed the whole reaction process. When there was only a little catalyst, under the same conditions, there would be only a few active sites of the catalyst, and the amount of H_2_O_2_ that can be activated is small, resulting in less amount of ·OH radical, which would lead to the lower removal of BPA. With the increase of catalyst dosage, the number of active sites provided by the catalyst increased, which was conducive to activation of a large number of H_2_O_2_ molecules, thus greatly increasing the content of **·**OH radicals in the solution, and realizing the degradation of BPA by heterogeneous CWPO in a very short time.

From Figure 7a, it can be seen that the removal of BPA was similar in the range of pH 3.0–7.9. After 65 min of reaction, the removal of BPA and the consumption of H_2_O_2_ both reached a high level under the conditions of pH 3.0, 4.7, 6.6, and 7.9. The removals of BPA were 94.0%, 91.3%, 87.6%, and 88.8%, respectively, and the consumptions of H_2_O_2_ were 97.0%, 99.9%, 99.9%, and 99.9%, respectively. The results showed that the catalytic effect of the catalyst was not greatly affected in the range of pH 3.0–7.9, which indicated that the catalyst had high pH adaptability. Because the pH value of BPA wastewater is usually about neutral, there is no need to adjust the pH value of wastewater in the reaction process.

From Figure 7b, it can be seen that under the conditions of pH values of 1.6, 2.0, and 3.0, after 85 min of reaction, the removals of BPA were 78.1%, 91.6%, and 98.1%, respectively, and the TOC removals were 69.5%, 81.5%, and 83.6%, respectively. It showed that the removal of BPA was still effective below pH 3.0, but the leaching of Cu^2+^ was serious. The leaching of Cu^2+^ was 106.8, 106.8, and 102.1 mg L^−1^, respectively. Ce^4+^ was not detected in the reaction solution, indicating that only Cu species were dissolved under these acidic conditions. With the pH values of 4.7, 6.6, and 7.9, the leaching of Cu^2+^ was 32.5, 26.4, and 23.9 mg L^−1^, respectively, and the leaching of Cu^2+^ was greatly reduced compared with the acidic condition. When pH was equal to 10.1, the leaching of Cu^2+^ reached the lowest value of 0.36 mg L^−1^, but the removal of BPA was only 8.2%. After that, the removal showed slight rebounds. This indicated that the strong acid environment caused great damage to the structure of CeCu oxide catalyst. When the pH value of the reaction system was relatively low, the BPA removal and TOC removal were lower in the early stage of the reaction, which could be attributed to the dissolution of CuO species in the catalyst under acidic conditions. As a result, there was a reduction in the number of active sites on the surface of the catalyst, and a hindering of the activation of the H_2_O_2_ molecule. However, as the reaction proceeded, the effect of pH decreased gradually. When the pH value of the reaction system was greater than 11.9, the removal of BPA decreased sharply, which could be attributed to the reaction of H_2_O_2_ with **·**OH to form HO_2_· and O_2_. It resulted in a sharp decrease in the amount of **·**OH produced in the solution and was not conducive to the oxidation conversion of BPA. It can be seen from the information above that CeCu oxide catalyst has strong pH adaptability, which may be closely related to the addition of rare earth element Ce to a large extent.

### 3.3. Stability of the CC450 Catalyst

Figure 8 shows that both the removal of BPA and TOC decreased with the increase of reuse times. After five reactions, the removal of BPA and TOC decreased from 90.5% and 80.5% to 86.7% and 76.2%, respectively. Figure 8 shows that after the first reaction, the leaching concentration of Cu^2+^ reached the highest, namely, 30.3 mg L^−1^. With the increase of reaction times, the leaching of Cu^2+^ decreased, and after five reactions, it was 28.9 mg L^−1^. The reasons for these changes may be as follows: firstly, with repeated reactions, the structure of the catalyst was damaged to a certain extent, and some amorphous CuO on the surface of the catalyst may have fallen off and dissolved in the wastewater, thereby losing some active components. Secondly, there were Lewis acid active sites [33] on the Cu species and CeO_2_ in the prepared catalyst where they could adsorb and catalyze organic compounds. However, the transition metal Cu^2+^ has an empty *d* orbit, which has strong electron absorption capacity [34]. At the same time, during the degradation of BPA, electron-rich functional groups with unsaturated bonds and aromatic rings may be formed [35], such as phenolic hydroxyl and carboxyl groups, which can easily form metal organic complexes or polymers with Cu species, resulting in some acidic active centers being covered by organic compounds, thus reducing the catalytic activity [36].

With the increase of the cycle times, although the catalytic oxidative degradation had a certain decrease, the removal of BPA was still maintained above 86.7%. The leaching concentration of Cu^2+^ gradually decreased, indicating that the prepared catalyst had a certain stability in the removal of BPA in heterogeneous CWPO.

### 3.4. Hydroxyl Radical and pH Changes in the Reaction

Figure 9 shows that in heterogeneous CWPO reaction, the change of pH value increased first and then decreased slightly. The results show that the alkaline environment formed rapidly in the initial stage of the reaction. With the prolongation of reaction time, acidic intermediate may be formed, which would make the pH value of the system decrease slightly, but the total pH value would remain greater than 8.1. The reaction system in alkaline environment can avoid the dissolution and destruction of H^+^ on the catalyst of CeCu oxide in acidic environment, which is conducive to the stability of the catalyst structure. Figure 9 shows that the concentration of **·**OH radical in CWPO reaction system decreased gradually with the prolongation of reaction time, indicating that a large number of OH radicals participated in the reaction. 

### 3.5. UV-VIS Spectra Change in CWPO Reaction System

BPA may contain molecular structures that can promote color generation. When connected with chromophores, the chromogenic ability of chromophores would be enhanced, causing red shift, namely, making the absorption wavelength shift to the long wavelength direction. By observing the color change of the reaction process, it was found that different colors produced in the initial stage of the reaction all occurred within 15 min, indicating that the color-promoting molecular reacted with chromophores during the reaction process. With the prolongation of the reaction time, these colored products were further oxidized and decomposed, with the colors disappearing. Figure 10 shows that after 25 min of reaction, the absorption peak shifted to the short wavelength direction, that is, the blue shift. It was indicated that the benzene ring structure in the BPA structure was prone to *π–π* transition or *n–π* transition, resulting in the destruction of the structure and the color-promoting group and the disappearance of color. Figure 10 shows that BPA produced a weak absorption band near 250 nm, which is generally referred to as a fine structure absorption band of an aromatic compound. It is also called a *β* absorption band, which is caused by a *π–π* transition and a benzene ring vibration. Thus, trace aromatic compounds may be formed during the degradation of BPA. As the catalytic oxidation proceeded, the intermediate products were further oxidized and decomposed. The ultraviolet-visible absorption peaks were further weakened or smoothed. Carboxylic acids such as formic acid, oxalic acid, and acetic acid may be formed or eventually oxidized to CO_2_ and H_2_O, and the absorption peaks disappear.

### 3.6. Catalytic Mechanism and Degradation Pathway

The results show that the degradation intermediates of BPA could hardly be detected when the sampling time exceeded 65 min. This may have been due to the strong oxidation ability of **·**OH radicals in heterogeneous CWPO reaction system, which can oxidize organic matter into small molecules or inorganic substances in a short time. The detection conditions of BPA intermediates were as follows: initial concentration of BPA 152 mg L^−^^1^, catalyst dosage 1 g L^−^^1^, H_2_O_2_ dosage 196 mmol L^−^^1^, room temperature and sampling within 3–10 min. The intermediates and structures detected by GC-MS are shown in Table 3. No BPA was detected during the reaction, indicating that the structure of the BPA would be destroyed at an early stage. The intermediate products also imply that the C–C bond and the O–H bond connecting the two benzene rings were attacked in the early reaction process, and a series of compounds containing a benzene ring were formed. As the reaction progressed, the compounds were gradually oxidized and degraded into small molecules, and finally mineralized into CO_2_ and H_2_O.

The detected intermediates show many similarities to that detected by Wenjing Chen [37], indicating that BPA may have the same degradation pathway under different catalyst conditions. It may be mainly due to the fact that the symmetry of BPA structure and the attack positions of strong oxidizing OH radicals are basically the same. In heterogeneous CWPO, the degradation pathway of BPA is shown in Figure 11. When BPA was exposed to catalyst and hydrogen peroxide, the reaction proceeded rapidly. At room temperature, colors gradually came into being. However, BPA was not detected in the detected intermediate products, indicating that the C–C or O–H bonds in BPA were destroyed in a short time. It is possible that the C–C bond was directly broken to form phenol, which would be further oxidized to form p-diphenol. The benzoquinone is formed after dehydrogenation of p-diphenol, which may also be reversible. Another route may have been the formation of another macromolecular ketone compound, which would be further oxidized to 4-ethylbenzaldehyde and p-isopropylphenol, and then to p-xylene and styrene. Phenol may have been formed in both paths, and the phenol would be further oxidized to acetic acid, propionic acid, formaldehyde, acetaldehyde, or oxalic acid, and finally further oxidized to CO_2_ and H_2_O.

According to the degradation pathway, CeCu oxide catalysts in heterogeneous CWPO reaction system may have mainly produced strong oxidative ·OH radicals by catalyzing H_2_O_2_ to achieve oxidative degradation of organic compounds. It was found that the prepared CeCu oxide catalyst formed a CeCu oxide solid solution at high temperature, and some Cu^+^ and Cu^2+^ replaced Ce^4+^ in the CeO_2_ lattice, which destroyed the coordination balance of CeO_2_ lattice and formed a pair of Cu^2+^ and Ce^3+^/Ce^4+^ ions between CuO and CeO_2_ [30], resulting in oxygen vacancy. Oxygen vacancy may be beneficial to the adsorption and activation of H_2_O_2_ [25]. The porous structure of the catalyst surface increased the possibility of the reaction between organic matter and hydrogen peroxide on the catalyst surface, which led to the organic matter being adsorbed, captured, and degraded on the catalyst surface. The catalytic mechanism of the catalyst may have been as follows:

Cu^2+^ catalyzed the H_2_O_2_ on the surface of the catalyst to decompose it into HO_2_· (1), then Cu^+^ reacted with H_2_O_2_ to form ·OH radical (2), Cu^2+^ reacted with HO_2_·to form Cu^+^(3), Ce^4+^ reacted with Cu^2+^ to form Ce^3+^ and Cu^2+^ (4). Ce^3+^ was oxidized to Ce^4+^ by H_2_O_2_, and generated ·OH radical (5), and then self-consumption reaction occurred. After the reaction process was over, the above chain reaction was repeated again.
(1)$${\mathrm{Cu}}^{2+}+H_{2}O_{2}\to \to {\mathrm{Cu}}^{+}+H^{+}+{\mathrm{HO}}_{2},$$
(2)$${\mathrm{Cu}}^{+}+H_{2}O_{2}\to \to {{\mathrm{Cu}}^{2+}+\mathrm{HO}+\mathrm{OH}}^{-},$$
(3)$${\mathrm{Cu}}^{2+}+{\mathrm{HO}}_{2}\to \to {\mathrm{Cu}}^{+}+H^{+}+O_{2},$$
(4)$${\mathrm{Ce}}^{4+}+{\mathrm{Cu}}^{+}\to \to {\mathrm{Ce}}^{3+}+{\mathrm{Cu}}^{2+},$$
(5)$${\mathrm{Ce}}^{3+}+H_{2}O_{2}\to \to {{\mathrm{Ce}}^{4+}+\mathrm{HO}+\mathrm{OH}}^{-},$$
(6)$$H_{2}O_{2}+\mathrm{HO}\to \to {\mathrm{HO}}_{2}+H_{2}O.\;$$

## 4. Conclusions

The CC450 catalyst prepared by citric acid complex method had less elemental composition, a simple structure, and high catalytic activity. The removal of BPA in the heterogeneous CWPO reaction system showed that after the reaction of 65 min, the removal of BPA and TOC could reach 87.6% and 77.9%, respectively. The biggest advantage of the catalyst was its strong pH adaptability, having high removal effect on BPA in the range of pH 1.6–7.9. Therefore, there was no need to adjust the pH value of the solution in the reaction process, thus greatly reducing the difficulty of artificial operation. Even when reused five times, the catalyst can still maintain its original crystalline phase structure and surface morphology to a large extent, with the composition and valence of elements on the catalyst surface nearly intact. The removal of BPA remained above 86.7%, the concentration of Cu^2+^ leaching decreased gradually, and the structure of the catalyst gradually kept stable.

In heterogeneous CWPO reaction system, the change of pH value increased first and then decreased slightly, remaining higher than the initial pH value. The system was in alkaline environment, with the overall pH value being greater than 8.1, which was conducive to the stability of the catalyst structure. With the prolongation of reaction time, the concentration of ·OH radical in CWPO reaction system decreased gradually, and the ultraviolet-visible spectra of BPA shifted from the short peak shifts to the short wavelength direction, that is, the blue shift. The catalysts degraded BPA rapidly in CWPO reaction system. In a short time, the C–C bond or O–H bond in BPA was destroyed, and there may have been two main degradation paths of phenol and ketone. The catalyst was suitable for the treatment of wastewater containing BPA and with a wide range of pH fluctuation, which did not require an adjustment of the pH value of the reaction solution. Therefore, it had high catalytic activity, a simple structure, and low cost, thus showing great applicability. 

## Figures and Tables

**Figure 1 ijerph-16-04675-f001:**
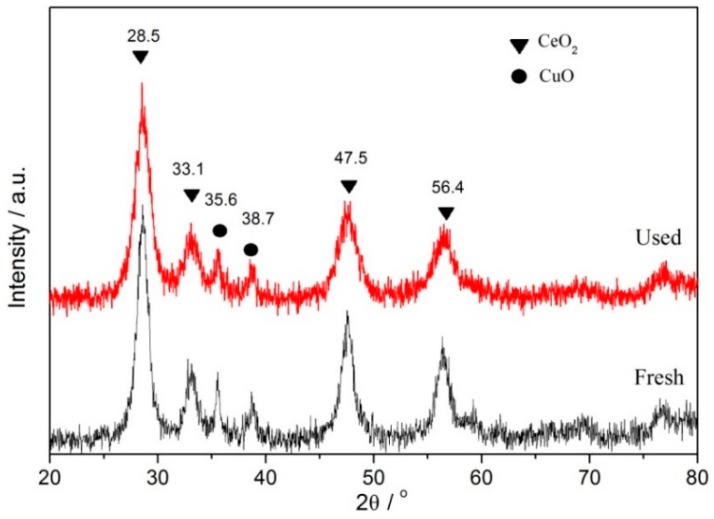
XRD patterns of fresh and used CC450 catalysts.

**Figure 2 ijerph-16-04675-f002:**
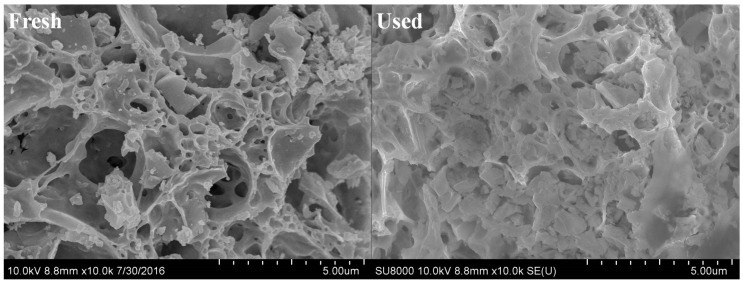
SEM photograph of fresh and used CC450 catalysts.

**Figure 3 ijerph-16-04675-f003:**
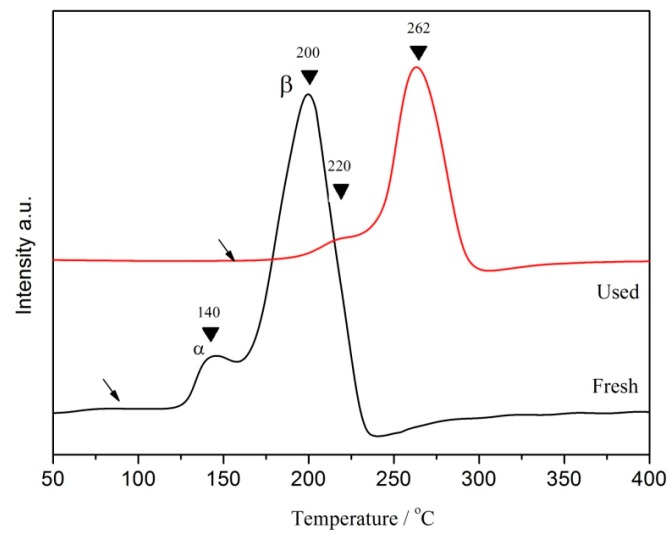
H_2_-TPR profiles of the fresh and used CC450 catalysts.

**Figure 4 ijerph-16-04675-f004:**
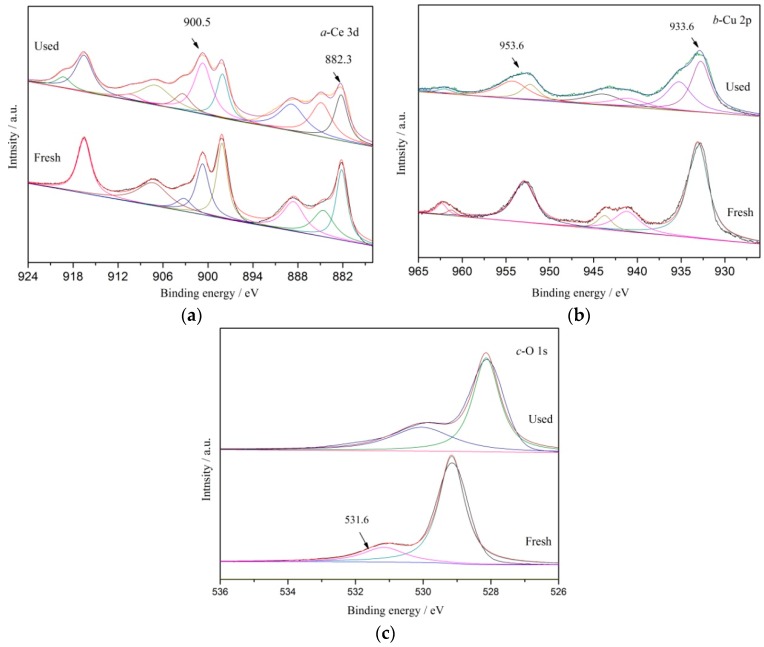
XPS spectra of fresh and used CC450 catalysts (**a**) Ce 3d; (**b**) Cu 2p; (**c**) and O 1s.

**Figure 5 ijerph-16-04675-f005:**
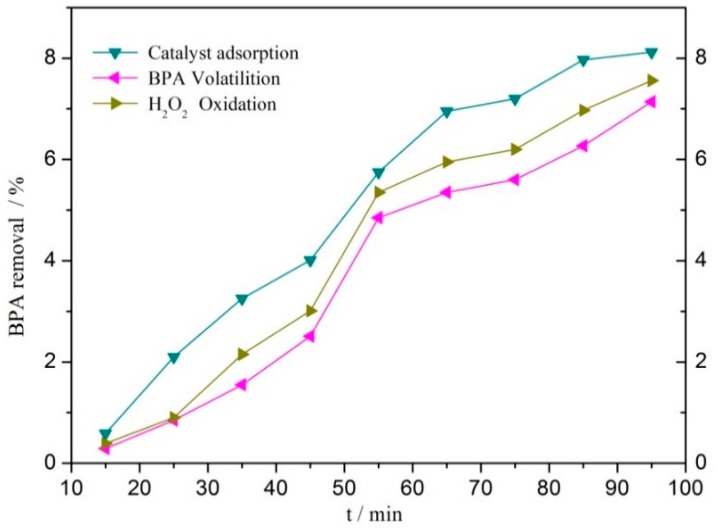
Effect of single factor on BPA degradation by catalytic wet peroxide oxidation (CWPO) with CC450 = 1 g L^−^^1^, H_2_O_2_ = 196 mmol L^−^^1^, BPA = 152 mg L^−^^1^, pH = 6.6, and *t* = 75 °C.

**Figure 6 ijerph-16-04675-f006:**
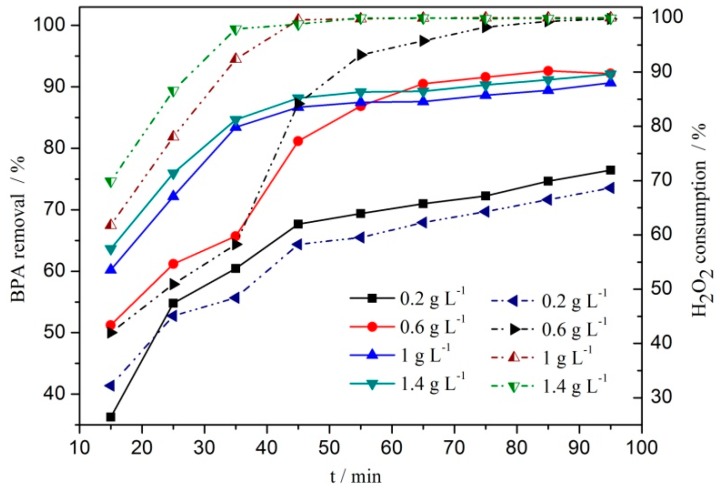
Effect of catalyst dosage on BPA degradation by CWPO with H_2_O_2_ = 196 mmol L^−1^, BPA = 152 mg L^−1^, pH = 6.6, and *t* = 75 °C (solid line, BPA; dashed, H_2_O_2_).

**Figure 7 ijerph-16-04675-f007:**
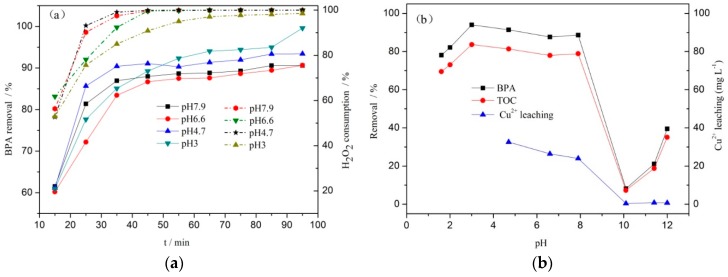
Effect of pH on BPA degradation by CWPO (**a**) with CC450 = 1 g L^−1^, H_2_O_2_ = 196 mmol L^−1^, BPA = 152 mg L^−1^, and *t* = 75 °C; (**b**) with CC450 = 1 g L^−1^, H_2_O_2_ = 196 mmol L^−1^, BPA = 152 mg L^−1^, and *t* = 75 °C, *t* = 85 min (solid line, BPA; dashed, H_2_O_2_).

**Figure 8 ijerph-16-04675-f008:**
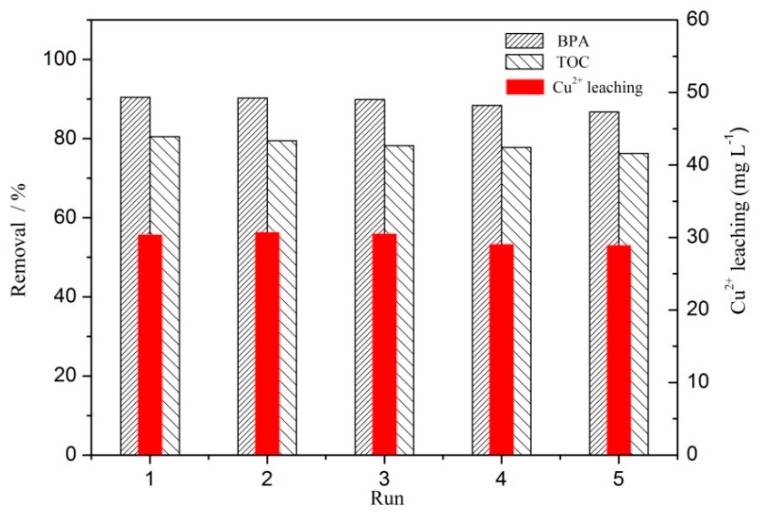
The reusability study of catalyst with CC450 = 1 g L^−1^, H_2_O_2_ = 196 mmol L^−1^, BPA = 152 mg L^−1^, *t* = 85 min, and *t* = 75 °C.

**Figure 9 ijerph-16-04675-f009:**
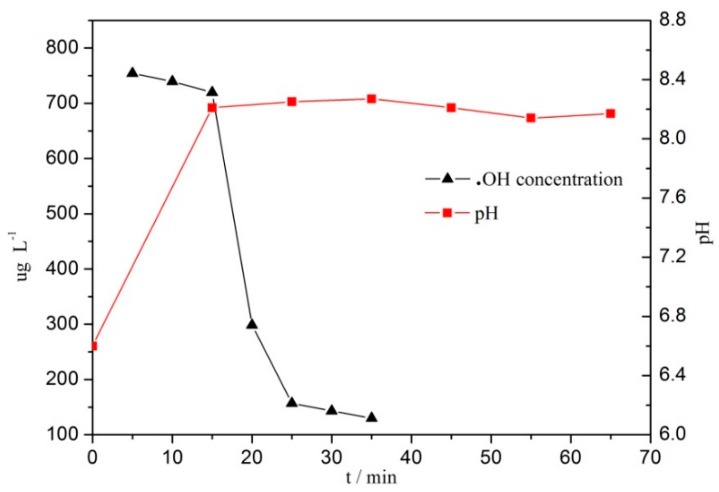
The change of pH and ·OH concentration in CWPO reaction system with CC450 = 1 g L^−1^, H_2_O_2_ = 196 mmol L^−1^, BPA = 152 mg L^−1^, and *t* = 75 °C.

**Figure 10 ijerph-16-04675-f010:**
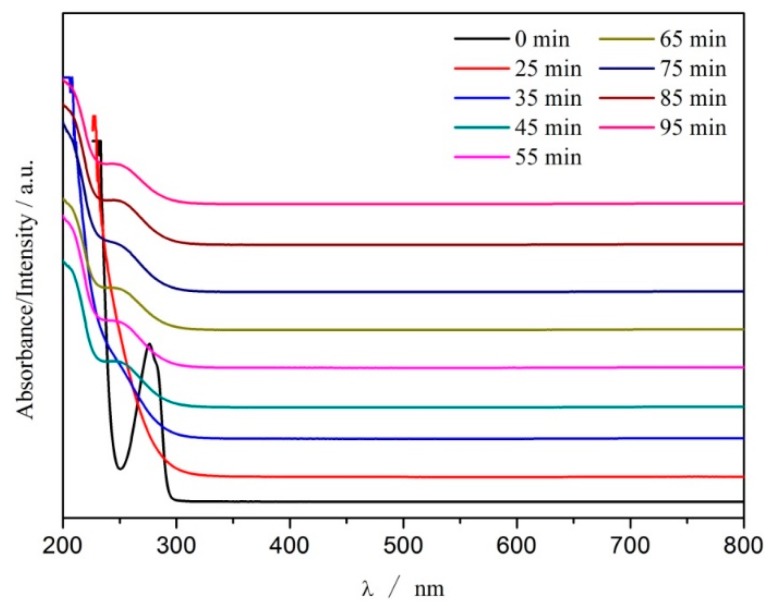
UV-VIS spectra of BPA in CWPO reaction system with CC450 = 1 g L^−1^, H_2_O_2_ = 196 mmol L^−1^, BPA = 152 mg L^−1^, and *t* = 75 °C.

**Figure 11 ijerph-16-04675-f011:**
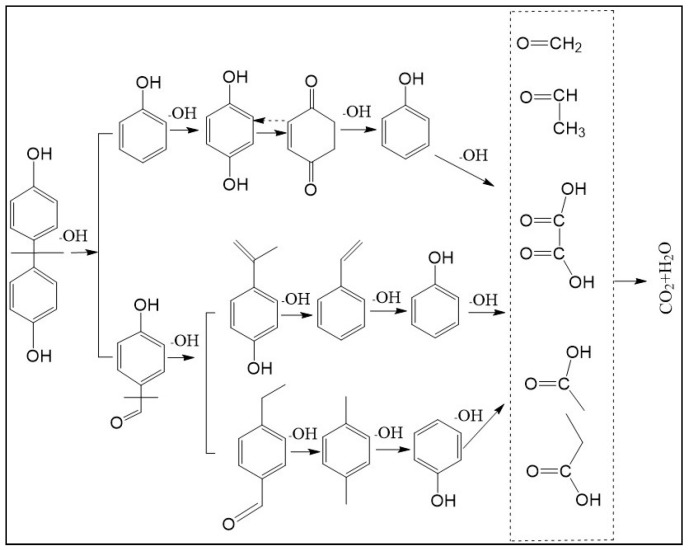
Probable degradation pathway of BPA.

**Table 1 ijerph-16-04675-t001:** Catalytic wet hydrogen peroxide oxidation of bisphenol A (BPA) by various heterogeneous catalysts. TOC: total organic carbon.

Catalyst	BPA or TOC Removal (%)	Conditions	Ref.
Fe-C (G S)	X_BPA_ = 100%, TOC = 60%	2.8% wt Fe, 100 mg L^−1^ BPA, 530 mg L^−1^ H_2_O_2_, pH = 3, 80 °C	[21]
Fe-Mt-TC-C (Iron-pillared montmorillonite Tetracycline carbon)	X_BPA_ = 100%, TOC = 78.3%	0.4 g L^−1^ Fe-Mt-TC-C, 0.4 mmol L^−1^ BPA, 15 mmol L^−1^ H_2_O_2_, pH = 3	[22]
Fe_3_O_4_-MWCNT (Multi-walled carbon nanotubes)	X_BPA_ = 97%	0.5 g L^−1^ Fe_3_O_4_-MWCNT, 0.3 mmol L^−1^ BPA, 1.2 mmol L^−1^ H_2_O_2_, pH = 3	[23]
Cu-TUD (Technische Universiteit Delft)-1	X_BPA_ = 90.4%	0.1 g 2.5wt% Cu/TUD-1, 100 ppm BPA, 90 mM H_2_O_2_, pH = 3.5	[24]
Cu-AlPO_4_	X_BPA_ = 88%	1 g L^−1^ Cu-AlPO_4_, 25 mg L^−1^ BPA, 10 mM H_2_O_2_, pH = 7	[25]
CuO-Al_2_O_3_	X_BPA_ = 100%, TOC = 91%	25 g L^−1^ CuO-Al_2_O_3_, 1 g L^−1^ BPA, 3.3 mL H_2_O_2_, pH = 2.4–7.2	[26]
CuFeO_2_	X_BPA_ = 100%, TOC = 85%	1 g L^−1^ CuFeO_2_, 0.1 mmol L^−1^ BPA, 20 mmol L^−1^ H_2_O_2_, pH = 5	[27]
(P, Cu, Ag, Fe, P, Cu)-Ti-PILC (Pillared clays)	X_BPA_ ≥ 87%	5 g L^−1^ m_cat_, 20 ppm BPA, 1360 ppm H_2_O_2_, pH = 4	[28]
PVP (polyvinylpyrrolidone)-AgNP	X_BPA_ = 95.5%	10 mg L^−1^ [AgNP]_0_, 0.4 mmol L^−1^ H_2_O_2_, pH = 4	[29]
Ag-AgCl-Fh	X_BPA_ = 100%, TOC = 92%	1 g L^−1^ Ag/AgCl/Fh, 30 mg L^−1^ BPA, 10 mM H_2_O_2_, pH = 3	[14]

**Table 2 ijerph-16-04675-t002:** Instrumental parameters for inductively coupled plasma optical emission spectroscopy (ICP-OES) measurements.

Parameter	Value
RF (Radio frequency) generator power (kW)	0.9
Frequency of RF generator/MHz	27.12
Nebulizer	Cross-flow
Plasma gas flow rate (L min^−1^)	12
Auxiliary gas flow rate (L min^−1^)	0.6
Nebulizer gas flow rate (L min^−1^)	0.6
Replicates	3
Integration time (s)	0.5
Wavelength (nm)	Cu, 224.70 nm Ce, 413.38 nm
Limit of detection (μg L^−1^)	1
Limit of quantification (μg L^−1^)	Cu, 3; Ce, 5
Repeatability	Relative standard deviation ≤ 1.5%
Stability	Relative standard deviation ≤ 2%
*R* ^2^	Cu, 0.99997; Ce, 0.99994

**Table 3 ijerph-16-04675-t003:** Possible major intermediates in the BPA degradation process.

Compound	Molecular Weight	Tentative Structure
Benzaldehyde, 4-ethyl-	134	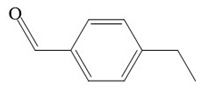
p-Benzoquinone	108	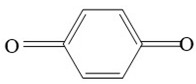
Phenol	94	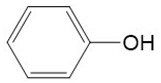
Styrene	104	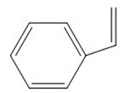
p-Isopropenylphenol	134	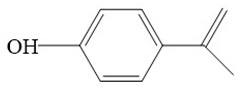
p-xylene	106	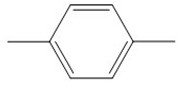

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
