# Peer review of "Degradation of Bisphenol A by CeCu Oxide Catalyst in Catalytic Wet Peroxide Oxidation: Efficiency, Stability, and Mechanism"

_ijerph, 2019, doi:10.3390/ijerph16234675_

Round 1
Reviewer 1 Report
The authors present the preparation CeCu oxide catalyst was prepared by citric acid complex method and the catalytic wet peroxide oxidation (CWPO) reaction system was established with Bisphenol A (BPA).
The subject is of interest, but the manuscript has many deficiencies, especially in the results.
Experimental
- Is necessary format homogenize of Hours and h, for example for 20 hours, and …… for 3.0 h, and space between the units, for example, 80°C ……100 °C
- The authors mention that the initial copper and cerium content of the prepared catalytic material and the leaching concentration of Cu2+ was measured by inductively coupled plasma optical emission spectroscopy …. In the results section, only there is for Cu2+ but no ICP results for the initial content of copper and cerium.
- In the Abstract, the authors mention that effects of catalyst dosage and pH on the removal efficiency of BPA were also investigated, but in the section 2.4., this description is not mentioned.
Results
- Discussion of TPR results is confusing
- Figures 6 and 7 are very sutured and are not understood.
- The results are presented with some inconsistencies and are little discussed. In general, all the section, the results are described in detail, but they are not analyzed or compared with the results of the literature.
References are double numbered, is necessary to correct
Author Response
Please see the attachmen.

Reviewer 2 Report
What is “ICP”? How to determine copper and cerium by the ICP analysis? Please describe the brand, model of the instrument and analytical procedures of ICP analysis.In general, BPA is quantified by GC/MS or LC/MS/MS. The authors used ultraviolet-visible spectrophotometer to determine BPA concentrations. How to confirm the QA and QC of the analytical method by ultraviolet-visible spectrophotometer (eg. accuracy..)?
The author used GC/MS to detect the intermediate products of BPA and BPA, respectively. Please describe the complete information of instrumental analysis (eg. the brand and model of GC column, GC and MS detector, and the parameters for GC inlet, oven…). Please also describe the LOQ, Qualitative and quantitative ions of BPA and the intermediate products of BPA. What is the “Qual” of the intermediate products of BPA, respectively? Did authors use standard solutions to re-confirm the intermediate products of BPA that found by the NIST library search?
Reviewer 3 Report
The paper by Jiao et Al. reports a thorough characterization of a CeCu supported catalyst for the degradation of organic (also aromatic) molecules in wet peroxide oxidation. As benchmark molecule, the authors employed the bisphenol A, which resulted completely mineralized at the end of the treatment (about 70 minutes). The authors characterized the material before and after the reaction, compared the results with a series of blank systems, studied the influence of the pH on the reaction, tested the stability of the catalyst in multiple sequential experiments and finally proposed a reaction mechanism consistent with experimental data.
In my opinion this paper certainly deserves to be published in IJERPH after some issues listed below will be addressed.
In Figure 1 is not clear whether the curve for the used material has been offset for the sake of clarity or the intensity is indeed higher than the curve for the fresh catalyst. Can the authors add the Intensity scale on the axis? At p.4 line 134, the authors write: “…the catalyst is less affected…”, less than what? Maybe the authors meant “slightly” instead of “less”? Please review this sentence. In Figure 6 the conversion of H2O2 reaches 100% after 30 min (1 and 1.4 g L-1 catalyst), but the conversion of BPA stops at 90%. Form these data it seems that H2O2 is the limiting reagent for the reaction. Did the authors try higher concentrations of H2O2? Moreover, the decomposition kinetics is quite different between BPA and H2O2, I was expecting more or less the same conversion rate for the two species. Can the authors comment on that? Finally, what happens before 15 min? Why data are not reported (this is true also for the other figures)? At p. 8 line 239 the leached concentrations of Cu2+ are reported as ~10000 mg L-1, I think it is a typo, please double check. What is the total volume of the solutions in the experiments (report this data in the experimental section)? 8 line 244, “mg L” is repeated twice. 8 lines 260 – 263, this paragraph is quite confuse and the meaning results obscure. Please rephrase it. 10 lines 293, “making” is repeated twice. 10 lines 291-298, the authors should replace “wave” with “wavelength”. 10 line 295, “…color-promoting…reaction process.” I can’t understand this sentence, please rephrase it. 13 reaction (5), a “+” sign is missing in the left side of the reaction. English is generally ok, but there are some typos and grammar mistakes along the text. Some of the sentences result obscure. Please, double check the whole text.
Reviewer 4 Report
Comments on the manuscript entitled “Degradation of bisphenol A by CeCu oxide Catalyst in Catalytic Wet Peroxide Oxidation: Efficiency, Stability and Mechanism” by Zhaojie Jiao et al. (ijerph-617000).
GENERAL COMMENT
The paper presents a group of interesting catalyst characterization tests and activity tests. This paper could be publishable, however the following comments and suggestions should be addressed in a revised manuscript before it can be reconsidered for publication.
(1) It is known that, during the catalytic oxidation of several compounds, polymeric compounds can be formed and promoted by acidic sites on the catalyst, which can be adsorbed at the surface of the catalyst and decrease its activity. Could you explain better this aspect with the results in the manuscript?.
(3) Could give the accuracy of your data in the figures 5 to 9?
(4) Could you precise your strategy, regarding the results obtained with this catalyst, to stop the loss of bet surface area and explain how the interaction of Metal with support could be increase the activity?
Other check:
(5) Please check the whole manuscript in order to correct English grammar mistakes.
(6) The references are not updated. It’s necessary some essential references.
(7) What is the novelty of this work?
(8) Why this pH?
This paper need a major revision and suggestions should be addressed in a revised manuscript before it can be reconsidered for publication.
Round 2
Reviewer 1 Report
The paper is better present and discuss, the authors have taken into account all the observations and improved the paper in general
Reviewer 4 Report
The authors have revised the manuscript according to the comments of reviewer.
